



# OMI UV aerosol index data analysis over the Arctic region for future data assimilation and climate forcing applications

Blake T. Sorenson[1], Jianglong Zhang[1], Jeffrey S. Reid[2], Peng Xian[2], Shawn Jaker[1]

[1]Department of Atmospheric Sciences, University of North Dakota, Grand Forks, North Dakota, 58202, United States of America
[2]Marine Meteorology Division, Naval Research Laboratory, Monterey, California, 93943, United States of America

*Correspondence to:* Blake T. Sorenson (blake.sorenson@und.edu)

**Abstract.** Due to a lack of high latitude ground-based and satellite-based data from traditional passive- and active-based measurements, the impact of aerosol particles on the Arctic region is one of the least understood factors contributing to recent Arctic sea ice changes. In this study, we investigated the feasibility of using the UV Aerosol
Index (AI) parameter from the Ozone Monitoring Instrument (OMI), a semi-quantitative aerosol parameter, for quantifying spatiotemporal changes in UV-absorbing aerosols over the Arctic region. We found that OMI AI data are affected by additional row anomaly that is unflagged by the OMI quality control flag and are systematically biased as functions of observing conditions, such as azimuth angle, and certain surface types over the Arctic region. Two methods were developed in this study for quality assuring the Arctic AI data. Using quality-controlled OMI AI data
from 2005 through 2020, we found decreases in UV-absorbing aerosols in the spring months (April and May) over much of the Arctic region and increases in UV-absorbing aerosols in the summer months (June, July, and August) over northern Russia and northern Canada. Additionally, we found significant increases in the frequency and size of UV-absorbing aerosol events across the Arctic and high Arctic (north of 80°N) regions for the latter half of the study period (2014-2020), driven primarily by a significant increase in boreal biomass-burning plume coverage.

## 1. Introduction

The Arctic region experienced noticeable changes in climate over the past two decades (Serreze and Francis, 2006; Serreze and Barry, 2011; Dai et al., 2019). Notable are the rapid melting of Arctic sea ice (Comiso, 2012; Dai et al., 2019; Kwok and Rothrock, 2009), increased permafrost melting (Kokelj et al., 2017; Blunden and Arndt, 2019; Liljedahl et al., 2016), and shifts in wildfire activity (Xian et al., 2022b). Despite being identified as a major factor
affecting the Arctic climate, atmospheric aerosol particles are still a large source of uncertainty in climate simulations (IPCC, 2013). Aerosol particles can alter Arctic climate directly through reflecting/absorbing solar incoming energy, absorbing terrestrial emission of IR radiation (for micron size particles such as dust), indirectly as cloud condensation nuclei by modifying cloud properties and increasing snow/ice melting through deposition of dust/smoke aerosols on snow- and ice-covered surfaces. All of these factors may very well interact between themselves and the overall Arctic
meteorology resulting in a difficult sea ice prediction problem.



One of the limitations of current Arctic aerosol studies is there are few space-borne measurements from traditionally aerosol-sensitive instruments (both passive- and active-based). This is largely due to the bright and variable lower boundary conditions of snow, ice and low clouds in the region (Martin, 2008). Consequently, there are no current operational aerosol retrievals that are available over the Arctic region from passive-based sensors such as Moderate Resolution Imaging Spectroradiometer (MODIS), Multi-angle Imaging SpectroRadiometer (MISR) and Visible Infrared Imaging Radiometer Suite (VIIRS) (Xian et al., 2022a). Active sensors, such as the Cloud-Aerosol Lidar with Orthogonal Polarization (CALIOP) on board the Cloud-Aerosol Lidar and Infrared Pathfinder Satellite Observation (CALIPSO) satellite, are able to provide retrievals of aerosol vertical profiles regardless of surface condition by measuring returned backscatter for the atmospheric layers below. Yet, CALIPSO's orbit only extends up to 82°N, missing a large portion of the Arctic region, and CALIOP aerosol retrievals suffer from "retrieval filled value" issue over or near the Arctic region due to the reduced sensitivity to optical thin aerosol layers (Toth et al., 2018).

The Ozone Monitoring Instrument (OMI), onboard the Aura satellite, is a nadir-viewing spectrometer that measures backscattered solar radiation at both ozone sensitive and non-sensitive channels (Levelt et al., 2006). The OMI aerosol index (AI), derived from the ratio of measured radiance to simulated radiance assuming Rayleigh atmosphere from an ozone sensitive channel, is able to detect UV-absorbing aerosols over bright surfaces such as clouds, deserts, snow, and ice (Torres et al. 2012; Alfaro-Contreras et al. 2014, 2016; Zhang et al., 2021). Launched in 2004, OMI provides one of the longest contiguous data records of the Arctic region. OMI AI suffers from its own problems over the Arctic region, including a well-known row anomaly issue that affects downstream products such as OMI AI (Torres et al., 2018). The OMI row anomaly first began in 2008 and is believed to be caused by a "physical obstruction", with the number of affected rows growing and decreasing over the years and now affecting over 30 rows, or over 50% of all OMI rows, and removing ~1/4 of coverage from each OMI swath. Further, long period average AI fields demonstrate an unnatural pattern of seasonal "rings." The ring, seen in the spring (April and May, Fig. 1a) and summer (June, July, and August, Fig. 1b), consists of high AI values in latitudes between approximately 70 °N and 80 °N and much lower AI values in latitudes north of approximately 80 °N. Additional high AI values are seen over shoreline regions in northern Russia, as well as along the ice-water boundary in the Greenland Sea.

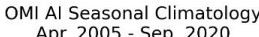

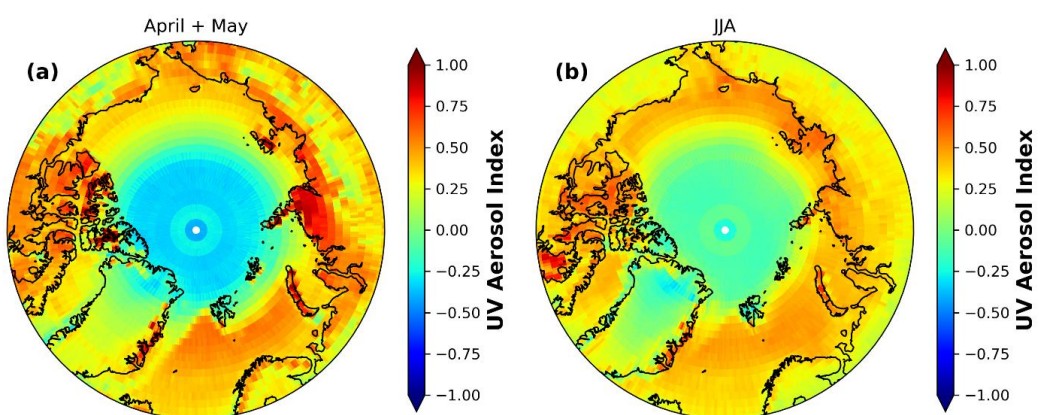

**Figure 1: Spring (April, May) and summer (June, July, and August) climatological averages of pre-QC OMI aerosol index (AI) between April 2005 and September 2020.**

In this study, we investigated uncertainties in OMI AI by enhancing this parameter's specificity by developing quality control methods. Using a revised and quality-controlled dataset, we studied extreme UV-absorbing aerosol events (dust and/or biomass burning smoke) over the Arctic region. Lastly, the developed OMI AI data may also be used for on-going OMI AI data assimilation efforts over the Arctic region (e.g. Zhang et al. 2021).

## 2. OMI datasets

Onboard the Aura satellite with a ~1:30 PM equatorial crossing time, OMI measures reflected solar energy between 270 – 500 nm at a nadir pixel size of 13 x 24 km (Levelt et al., 2006). Using radiance measurements at the 354 nm spectral channel, the OMI aerosol index is derived based on Eq. (1):

$$UVAI = -100 \log \left[ \frac{I_\lambda^{obs}}{I_\lambda^{cal}} \right], \tag{1}$$

where $I_\lambda^{obs}$ is the observed radiance and $I_\lambda^{cal}$ is the calculated radiance for a hypothetical pure Rayleigh scattering atmosphere. Over non-snow/ice surfaces, $I_\lambda^{cal}$ for the operational product is calculated by considering both clear and cloudy sky contributions, but over snow/ice surfaces, $I_\lambda^{cal}$ is calculated assuming a Lambertian surface reflectivity and with no consideration of cloud cover status (Torres and Leonard, 2018). OMI OMAERUV V003 UV aerosol index data from the Aura OMI level-2 near UV aerosol data product 'OMAERUV' are retrieved from the Goddard Earth Sciences Data and Information Services Center (GES DISC) archive for times between 01 April and 30 September each year from 2005 through 2020 (Torres, 2006). Sunlight is absent from the Arctic region during the boreal winter months, so only UVAI data between 01 April and 30 September of each year are analyzed.



### 3. Observed bias/uncertainties in OMI AI data

As the first phase of the study, uncertainties in OMI AI data, including row anomalies as well as uncertainties associated with observing conditions such as viewing geometries and surface properties are investigated near/over the Arctic region.

80   **3.1 Row anomaly**

The first possible cause for the AI ring over the Arctic region as shown in Fig. 2 may be associated with OMI row anomaly. In the OMI data, row anomalies are highlighted with a quality control flag named the XTrackQualityFlag (Xtrack). The Xtrack values change from zero to four, representing a row as "not affected" (Xtrack value of zero), "affected, not corrected, do not use" (Xtrack value of one), "slightly affected, not corrected, use with caution" (Xtrack value of two), "affected, corrected, use with caution" (Xtrack value of three), and "affected, corrected, use pixel"

85   (Xtrack value of four) by the row anomalies.

However, even after applying the Xtrack flag screening (by using OMI AI data with Xtrack = 0 only), additional bad sensor rows are found throughout the OMI data record (e.g. Fig. 2). As seen in Fig. 2a, which shows the OMI AI values for 10 April 2012, 21:52:00 UTC, two rows (43 and 44) with significantly high AI values of above three are

90   found in the middle of the swath, with the adjacent rows (45 – 50) show much lower AI. These same two rows report similarly high AI in the following swath at 10 April 2012, 23:30:00 UTC (Fig. 2b), indicating that the AI signal in the two swaths is non-meteorological and is caused by currently unflagged row anomaly. The unflagged anomalous rows in the OMI dataset, which seemingly latitude dependent, must be identified and removed from further analysis.



**Figure 2: a)** Single-swath OMI UV Aerosol Index data from the 10 April 2012, 21:51:00 UTC swath. The large gap in the middle of the swath is caused by the removal of flagged row anomaly-affected OMI data, while the red portion of the scan line over the Arctic is caused by unflagged row anomaly. **b)** Single swath OMI AI from 10 April 2012, 23:30:00 UTC. **c)** Averages of AI from each OMI sensor row over the Arctic for the 23:30 UTC 10 April 2012 swath. **d)** Single swath OMI AI from 10 April 2012, 23:30:00 UTC but after removing the anomalous rows 43 and 44 identified in c. **e)** Flagged (blue) and unflagged (red) row anomaly-affected OMI sensor rows not flagged by the XTrackQualityFlag variable in the OMI data files.



The first step in cleaning the OMI data is to remove the bad scan rows that are not flagged by the Xtrack flag through the entire study period of 1 April 2005 to 30 September 2020. Daily averages of AI from the northern end of all 60 OMI sensor rows over the Arctic are calculated, and if any one of those 60 row averages is more than two standard deviations away from the mean of all the 60 row averages, it is flagged as a bad row. For example, for the single OMI swath shown in Fig. 2b, the averages of the AI values from each row over the Arctic (Fig. 2c) reveal that the average AI in rows 43 and 44 are significantly higher than in the other rows, more than twice as large as any of the other row averages from the swath and nearly 400% higher than the nearby rows 47 and 48. Fig. 2e shows the 'flagged' (blue) and 'unflagged' (red) row anomaly-affected rows in the Arctic OMI data between 1 April 2005 and 30 September 2020. The flagged rows in the figure reflect any row in which at least one pixel over the Arctic has a non-zero Xtrack QC flag value, indicating that it is affected by the row anomaly. The unflagged rows are more than two standard deviations away from the average of all rows over the Arctic (indicating row anomaly), but are not flagged by the Xtrack QC flag. As shown in Fig. 2e, the bad rows identified by the algorithm are variable across the dataset time period, with scattered unidentified bad rows found in the 10s before 2012 and others in the 40s found between 2012 and 2013. The most strongly affected unflagged rows found by the algorithm are rows 24, 22, and 53, with row 24 being contaminated from 2013 to 2015, row 22 being contaminated in 2016, and row 53 being contaminated from 2016 until at least the end of the time period. The unflagged bad rows found for each day are used in further analysis to pre-screen the AI data before applying the main QC methods.

### 3.2 Other observing condition related uncertainties

Interestingly, we found that AI patterns are highly dependent upon observing conditions such as surface properties and viewing geometry, likely associated with the retrieval algorithm. This can be illustrated by evaluating AI patterns over the same region for similar observing conditions but with observations separated by almost exactly one year. For example, the OMI swaths from 02 April 2007, 00:51:00 UTC (Fig. 3a) and one year later on 02 April 2008, 00:57:00 UTC (Fig. 3b) exhibit nearly identical AI patterns along the coast of northern Russia. Clearly, the repeated patterns in OMI AI indicate that they are systematic, and are associated with surface properties and viewing geometries as mentioned below. Despite this observing condition dependency, aerosol events can still be detected using OMI AI data as shown in Fig. 3c and Fig. 3d. Figure 3c shows the OMI swath from 22 April 2007, 15:19:00 UTC and Fig. 3d shows the OMI swath from almost one year later on 22 April 2008, 15:24:00 UTC (Fig. 3d). While similar patterns of moderate AI are observed along the northern Canadian and Alaskan coasts, the smoke plume extending over northern Alaska is still detectable in Fig. 3d. Figure 3 reveals that OMI AI data can still be used in aerosol study over the Arctic region, but there are systematic biases in the OMI AI data that must be considered before using the data for scientific applications.

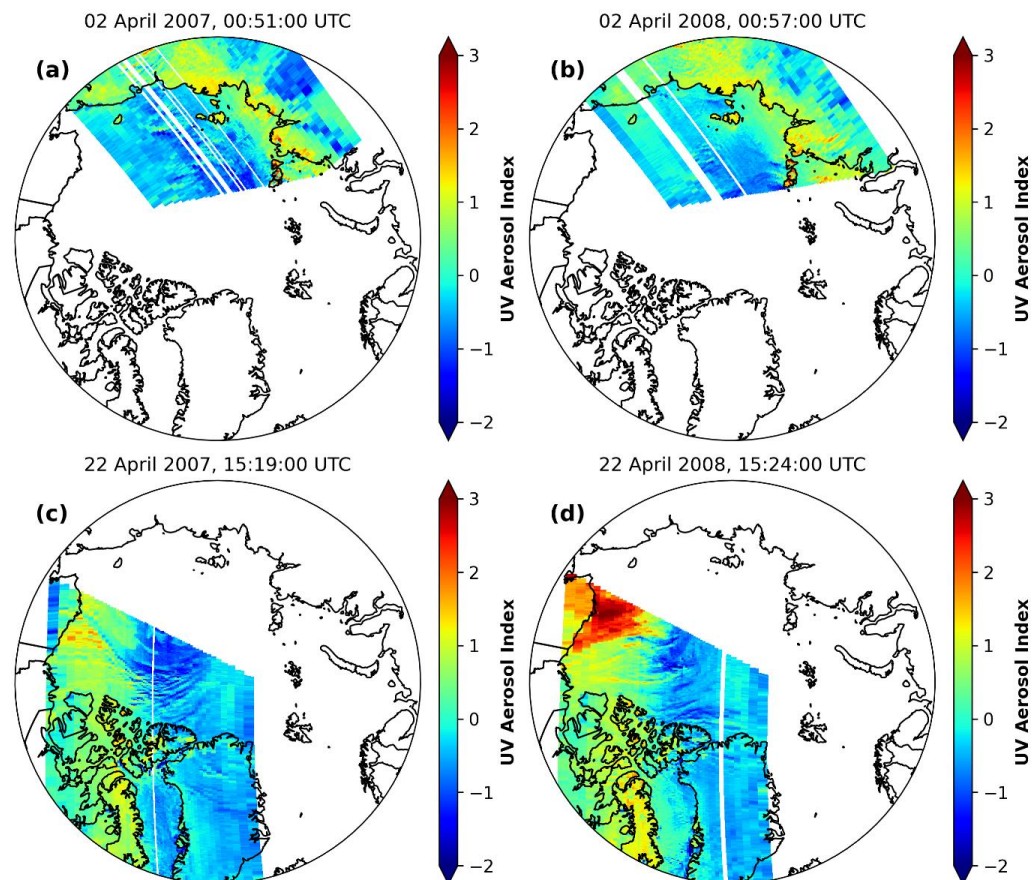

**Figure 3: Single swath OMI AI data from a) 02 April 2007, 00:51:00 UTC b) 02 April 2008, 00:57:00 UTC c) 22 April 2007, 15:19:00 UTC, d) 22 April 2008, 15:19:00 UTC.**

One of the causes for the systematic bias in OMI AI as seen in Fig. 3 is related to surface properties, with anomalously high AI values are found in association with certain surface types. To examine the impact of surface properties on anomalies in OMI AI, we use the GroundPixelQualityFlag (GPQF), which is included in the OMAERUV data. Each GPQF variable is a 16-byte unsigned integer and different bit ranges are used to store different characterizations. Bits 0 – 3 contain the land/water flags, including "shallow ocean", "land", "shallow inland water", and "deep inland water". The bits of interest for studying the isolated high AI values are bits 8 – 14, which contain the snow/ice flags. The flag values (Table 1) contain flags for snow-free land, sea ice concentration from 1% to 100%, permanent ice (used mostly for Greenland and Antarctica), dry snow, and ocean, among others.

**Table 1. OMAERUV snow/ice flags, taken from bits 8 – 14 of the GroundPixelQualityFlags found in each OMI data file. This table is adapted from information described in the OMI File Specification Document (Ahn et al., 2011).**

| Bit value | Flag |
|---|---|
| 0 | Snow-free land |





| 1-100 | Sea ice concentration (percent) |
|---|---|
| 101 | Permanent ice (Greenland, Antarctica) |
| 102 | Not used |
| 103 | Dry snow |
| 104 | Ocean (NISE-255) |
| 105 – 123 | Reserved for future use |
| 124 | Mixed pixels at coastline (NISE-252) |
| 125 | Suspect ice value (NISE-253) |
| 126 | Corners undefined (NISE-254) |
| 127 | Error |

Anomalously high OMI AI values are found to be associated with the "dry snow" surface class for high latitudes, which denotes regions covered in seasonal snow, unlike the "permanent ice" flag, which denotes regions that are assumed to be covered with snow year-round (Stammes and Noordhoek, 2002). For example, the OMI AI data from

the 22 April 2008, 10:27:00 UTC swath (Fig. 4a) show isolated regions around Greenland and the Canadian Arctic Archipelago with AI values of at least 2, much higher than the surrounding areas. The GPQF surface type classification values for the same swath (Fig. 4b) show that much of Greenland and the northeastern Canadian Arctic Archipelago are classified as "permanent ice" (seen as the cyan color in Fig. 4b), but there are also some areas classified as "dry snow". The isolated areas of "dry snow" match up well with both the areas of isolated high AI in the single-swath AI

data and the isolated climatologically high AI seen in the same regions in Fig. 4a. The isolated, anomalously high UVAI values in the Canadian Arctic Archipelago are found in the same places as the pixels classified as 'Dry Snow/Ice' in the GroundPixelQualityFlags. This suggests that different algorithms are being used in the UVAI calculations between the two surface classification types. As mentioned in the data section, different algorithms are applied over non-icy regions versus snow- and ice-covered regions (Torres and Leonard, 2018).

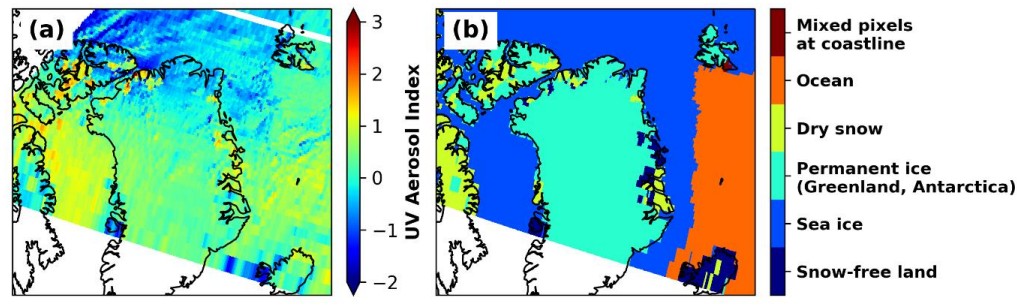


**Figure 4:** Pre-QC OMI UVAI (a) and the land/water flag values extracted from the OMI GroundPixelQualityFlags (b) for the 22 April 2008, 10:27:00 UTC OMI swath.

Another cause for the systematic bias in OMI AI is linked to azimuth angle or the row number. For example, Fig. 5a shows the average of all OMI AI data from each row over the Arctic between 01 April 2006 and 30 September 2006





(Fig. 5a, blue) and the relative azimuth angles for each OMI row (Fig. 5a, orange). OMI rows 1 to 30 have a relative azimuth angle of about 70° and OMI rows 31 to 60 have a relative azimuth angle of about 110°. Not only is the average AI in rows 1 to 30 much higher than the average AI in rows 31 to 60, but the average AI in rows 1 to 30 varies significantly as a function of row number. In contrast, the average AI in rows 31 to 60 do not vary as a function of row number and remain at about an average AI value of 0, with slight variation. This bias can be seen in OMI AI data

from two aerosol-free swaths on 22 April 2006: one at 10:50:00 UTC (Fig. 5b) and another from two swaths later at 14:08:00 UTC (Fig. 5c). The OMI data from the 10:50:00 UTC swath over Greenland are sampled using the lower 30 scan lines and exhibit AI values near 1, which, by the definition of OMI AI, indicates the presence of UV absorbing aerosols. Yet, large amounts of UV absorbing aerosols are normally not expected over Greenland for this season (e.g. Xian et al., 2022b). The same region viewed with the higher 30 scan lines two swaths later exhibits much lower AI

values below 0, indicating this region is free of UV-absorbing aerosols. Similar patterns can be routinely observed, with abnormal OMI AI values found for observations with row numbers of 1-30 (or relative azimuth angle of below 100°). This suggests that systematic biases exist in OMI AI data associated with either row number or relative azimuth angle, and higher than normal OMI AI values are found for observations with relative azimuth angle lower than 100° or row numbers lower than 31.

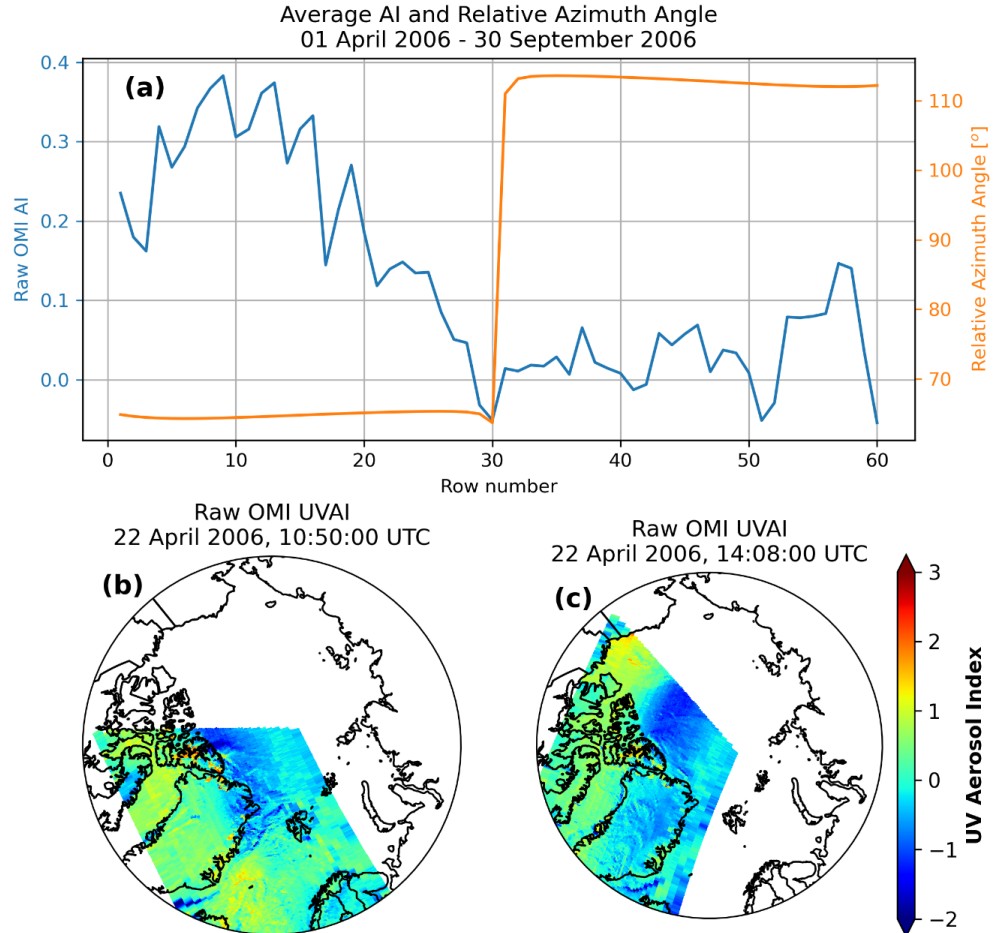


**Figure 5: a) Row-based climatology of OMI AI over the Arctic region calculated between 01 April 2006 and 30 September 2006 (blue) and the relative azimuth angle associated with each OMI sensor row from the 22 April 2006, 10:50:00 UTC OMI swath (orange). b) OMI AI data from the 22 April 2006, 10:50:00 UTC OMI swath. c) OMI AI data from the 22 April 2006, 14:08:00 UTC swath, two OMI swaths after the swath shown in panel b.**

**4. Methods**

**4.1 Data QC methods**

Knowing the issues in OMI AI data over the Arctic region, two different methods are presented for quality control of OMI AI data. In this first method, which is referred as the 'screening method', all non-reliable data are removed. As discussed in the previous section, abnormally high OMI AI values are associated with certain surface types and low

relative azimuth angles. Thus, AI pixels with relative azimuth angle less than 100º, ground classification type of 'dry snow', and either flagged (Xtrack flag not equal to 0) or unflagged (as identified above) row anomaly are excluded. For the climatological study, we also used only rows 56-60, as we found that only rows 56-60 have high relative





azimuth angle larger than 100° and are unaffected by row anomaly through the entire study period (2005 – 2020). The advantage to the "screening method" is that unperturbed and quality assured OMI AI data are included. Also, with the

use of the same set of rows (56-60), sampling bias is reduced. Still, due to the stringent selection criteria, only a small fraction of the data over the Arctic region pass this quality check. For example, for daily averages of AI for 17 July 2018 using only rows 56 through 60, only 12.2% of all quarter-degree lat/lon boxes north of 65° N have data, compared to 51.1% when all good rows are used. When all good rows are used, about 85% of quarter-degree grid boxes between 70° N and 80° N have daily observation coverage, but that coverage drops to about 6% when only rows 56 through 60

are used. North of 80° N, only about 20% of grid boxes have observation coverage in both methods because rows 56 through 60 are the only functional rows that sample in that region.

As discussed in the previous section and as shown in Fig. 4, OMI AI biases are strong and systematic functions of viewing geometries and surface conditions is linked to azimuth angle or the row number. Thus, in the second method, systematic patterns in OMI AI as functions of surface properties and viewing geometry are constructed using 15 years

of OMI AI data over the Arctic region. Then, by excluding those systematic patterns, perturbations in OMI AI values can be derived and further used to study spatiotemporal trends of OMI AI over the region. This method is called the "perturbing method". For this method, an OMI AI clear sky climatology is constructed as a function of viewing geometry and ground classification. As with the screening method, all bad rows from each day (both flagged and unflagged) are removed. Then, each OMI pixel from every swath in the 15-year dataset used in this study is binned

by solar zenith angle (SZA), viewing zenith angle (VZA), relative azimuth angle (AZM), spectral albedos from two channels (ALB1 and ALB2), and surface type (SFCT). The addition of the SFCT dimension allows the removal of AI data associated with faulty surface types, such as data over the 'dry snow' regions in the Arctic. Thus, for a given set of observing conditions, SZA, VZA, AZM, ALB1, ALB2, and SFCT values of each original OMI pixel are used to compute climatological AI values. Perturbations in OMI AI due to unrealized aerosol plumes are therefore identified.

Figure 6 shows an example of the results of applying both the screening and perturbing methods to the OMI swath from 22 April 2008, 21:59:00 UTC. While a large smoke plume over Alaska and the Arctic Ocean is seen in the raw, pre-QC AI data (Fig. 6a), the pre-QC data also exhibit significant bias in AI across the sensor rows, with the lower scan lines near the Siberian coast having generally higher AI than the higher scan lines over the Arctic. After applying the screening method to the AI data, the screened data (Fig. 6b) retain the AI signal north of Alaska while removing

the biased rows with azimuth angle less than 100°; however, the data volume is significantly reduced. Figure 6c shows the binned climatological values associated with the SZA, VZA, AZM, ALB1, ALB2, and SFCT values in each pixel from the swath. The climatological values reveal the row bias seen in the lower scan lines as well as the SFCT-induced bias over the coastal regions in northern Russia. Possible row anomaly effects are also seen in the mid-range scan lines, with several rows in the middle of the Arctic Ocean having slightly higher AI than the nearby rows. Figure 6d

shows the perturbed AI values, calculated by subtracting the climatological AI values shown in Fig. 6c from the raw AI values shown in Fig. 6a. The AI signal over Alaska, as well as weak signal from the Siberian coast, is retained while the row- and SFCT-induced biases are removed. One downside of this approach is that the resulting cleaned dataset consists of AI perturbations. Thus, this approach is better suited to identifying the seasonal behavior and frequency of Arctic aerosol plumes.

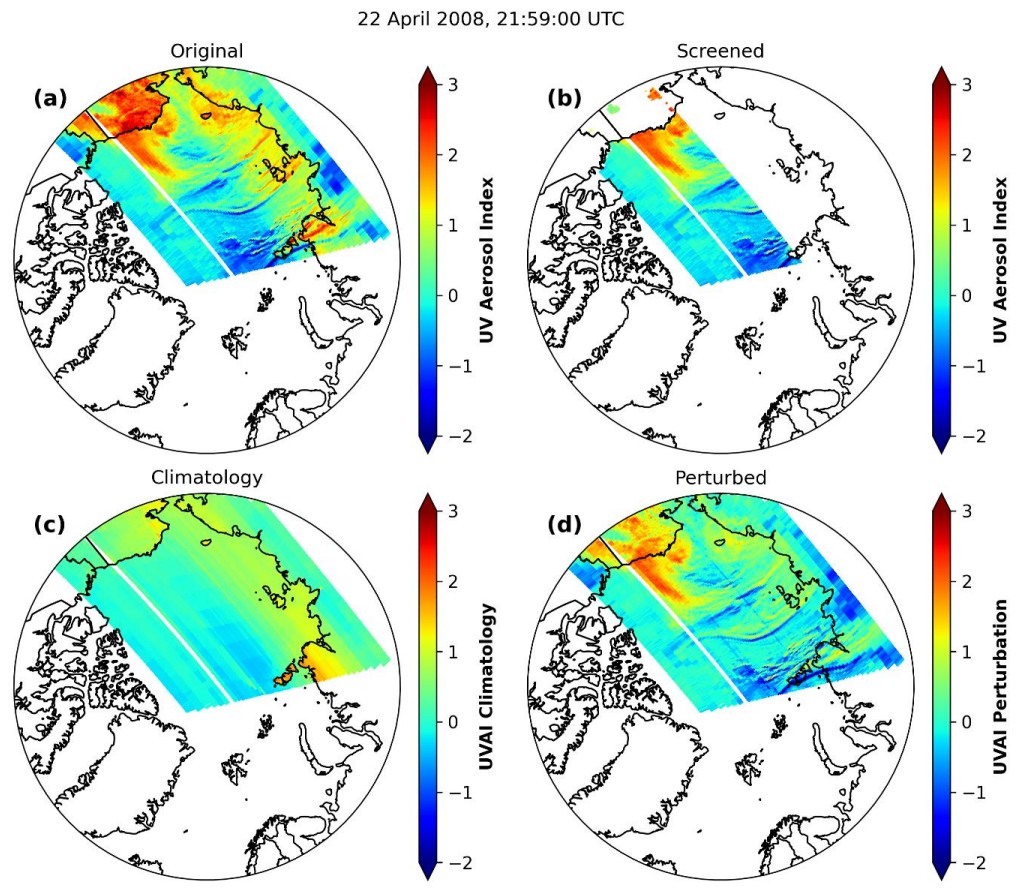

**Figure 6: Results of applying the screening and perturbating QC methods to an OMI swath from 22 April 2008, 21:59:00 UTC. a) the raw, pre-QC OMI AI data. b) the screened OMI AI data. c) the binned climatological values associated with the SZA, VZA, AZM, ALB1, ALB2, and ground type values of each pixel from this swath. c) the cleaned, perturbed AI data calculated after subtracting the climatology values from the pre-QC AI values.**

### 4.2 OMI sensor drift check

It is necessary to explore potential signal drift and signal degradation in OMI AI data for the trend analysis. To determine if any signal drift is present in the OMI AI dataset, monthly averages of AI are calculated for a remote ocean region (0 – 40 S, 180 W – 140 W) using the screening approach and with daily bad rows removed, as identified by the bad row algorithm; total observation counts in the region for each month are tracked as well. The remote ocean region is used as this region is assumed to be free from major aerosol pollutions (e.g., Zhang and Reid 2010). To reduce sampling bias, only data from rows 56 through 60 are included in this analysis. The observation counts (Fig. 7, orange) decrease slightly from 2005 to 2012, with local minima in counts in 2009 and 2011 which reveals that at least one of the 6 rows used in the second analysis was contaminated for a short amount of time between 2009 and 2011; this confirms the row anomaly timeline reported by Torres et al. (2018). The average AI (Fig. 7, blue) shows a



similar dip between 2009 and 2011, but like the observation counts, returns to the original average value of about 0.3

by 2012 and remains at about the same value until 2020, when a slight increase is found. No obvious signal drift is

found in the data as shown in Fig. 7 between 2008 and 2019.

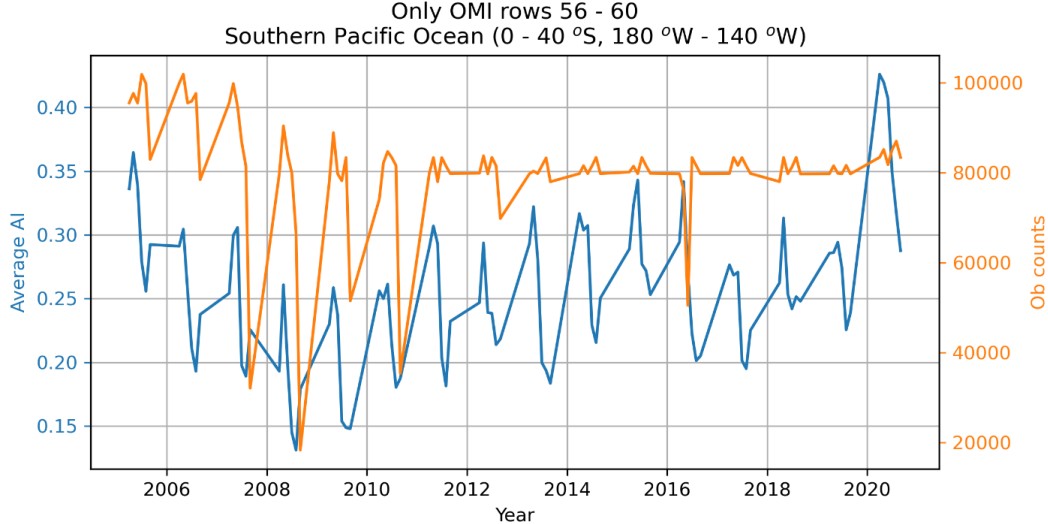

**Figure 7: Monthly average AI (blue) and total observation counts (orange) in a remote-ocean region in the southern Pacific
Ocean (0 – 40 °S, 180 W – 140 °W), calculated using the screening criteria and removing bad rows identified by the bad
row detection algorithm.**

### 4.3  OMI trend analysis

Trends are calculated for both the screened and the perturbed OMI AI data. Monthly averages of both the screened

and perturbed OMI data are first calculated on a 1x1 degree lat/lon grid. Then, monthly trends are calculated at each

lat/lon grid point by performing linear regression on all averages from the month being analyzed; for example, if the

May monthly trend is being calculated for a grid point, linear regression is applied to fit a line to the monthly averages

from May of every year from 2005 through 2020. After the regression line is fitted to the data, the slope of the trend

line is multiplied by the number of years in the study period to determine the AI trend over the study period. These

methods are applied for each grid box for all six monthly trends (April, May, June, July, August, and September).

Trend significance is calculated using a Wald t-test (Wald, 1943), and trends are considered statistically significant if

the p value from the Wald slope hypothesis test is less than 0.05, which signifies significance at the 95% confidence

level.





## 5. Arctic OMI AI climatology, trend, and extreme event statistics

**5.1 Monthly climatology and trend of Arctic OMI AI**

The screened and perturbed OMI data are applied to Arctic AI monthly summer climatology and trends between 2005 and 2020, as shown in Fig. 8. The first column in Fig. 8 shows the April, May, June, July, August, and September monthly AI climatology calculated without applying either of the QC methods, with only the original row anomaly check (Xtrack flag equal to 0) applied. Considerable ring affect is found in all six monthly climatologies, with the

strongest ring effect found in August and September. The second column shows the same climatologies calculated using the screened OMI AI data. Not only is the overall climatological average significantly reduced, but most of the AI ring found in the original climatologies is removed.



**Figure 8: April (first and top row), May (second row), June (third row), July (fourth row), August (fifth row), and September (sixth and bottom row) monthly climatologies of un-QC-ed OMI AI data (first column) and screened AI data (second column), as well as trends in the un-QC-ed AI data (third column), the screened AI data (fourth column), and the perturbed AI data (fifth column). Climatology and trend are calculated between 2005 and 2020. The dotted regions in the right two columns denote trends that are statistically significant at the 95% confidence level.**






The monthly trends calculated using the pre-QC OMI data, shown in the third column in Fig. 8, are very noisy, with
overall positive AI trend found across nearly the entire Arctic in all analyzed months, as well as a ring of strong
positive trend found north of approximately 80° N. However, after applying both the screening (Fig. 8, fourth column)
and perturbing (Fig. 8, fifth column) methods to the AI data, the overall positive AI trend is removed, with widespread
statistically significant negative AI trends over the Arctic region found in the April and May monthly trends. The June
and July monthly trends reveal increasing AI over northeastern Russia and Alaska, with the positive AI trends over
Russia being statistically significant, while the August trends reveal increasing AI over north-central Russia and
northern Canada, with the Russian positive AI trends being statistically significant. These results agree with AOD
trend statistics simulated by chemical transport models assisted with MODIS, MISR, and CALIOP data analysis by
Xian et al., who found decreasing Arctic region AOD in the spring months and increasing Arctic region AOD in the
summer months (Xian et al., 2022a).

Over the Arctic Ocean, mostly negative AI trends are found in June and July, with some areas of increasing AI trend
found over the Chukchi Sea (northwest of Alaska) in the July trends. A disagreement between the screened and
perturbed trends exists in June and July, with the perturbed trends reporting a half-circle of positive AI in the Kara
and Greenland Seas (north of Norway and northwest Russia) that is not found in the screened trends. One possible
cause for this discrepancy is a series of UV-absorbing aerosol events in the latter part of the study period over the
Kara and Greenland Seas that were fast-moving, short-lived, and did not extend very far north. Under these conditions,
OMI rows 1 – 22, which are only included in the perturbed data, would have observed the plumes while OMI rows
56 – 60, which are used in both methods, would not have observed the plumes, leading to the screened trends missing
the effects of said plumes. The consistently negative AI trends over the Greenland land mass for all months appear
non-meteorological, as these negative trends are surrounded on all sides by positive AI trends over the ocean water.
300   Due to the stark difference in trend between the oceanic and land-based trends over Greenland, and the
climatologically low AI over Greenland found in the screened AI climatology found in Fig. 8, the negative AI trends
over Greenland are suspected to be caused by lower boundary condition issues and are not meteorological.

## 5.2  Arctic smoke plume frequency analysis

The perturbed AI dataset allows for a unique study of the frequency of aerosol events over the Arctic (70° N – 80° N)
305   and high Arctic (north of 80° N) that cannot be provided by other sensors. To study the Arctic aerosol event frequency,
the perturbed AI dataset is averaged into daily quarter-degree lat/lon grids, and the total area of the quarter-degree
grid boxes with average perturbed AI beyond a threshold value is calculated for each day between 1 April 2005 and
30 September 2020. For this study, a perturbed AI threshold value of 1.0 is used to remove any residual non-
meteorological noise from the data (note that results do not change significantly if the threshold is changed to a higher
value such as 1.5 or 2.0). Figure 9a shows the time series of daily total area of quarter-degree lat/lon grid boxes with
perturbed AI greater than 1.0 between 70° N and 80° N. Most Arctic aerosol events occur in July and August, with
June having the next most aerosol events. Cool-colored lines in the figure indicate events occurring in the early
portions of the study period (2005 – 2011) while warm-colored lines indicate events in the later portions of the study
period. As shown, most of the large aerosol events between 2005 and 2020 occurred within the later portions of the





study period. Additionally, this time series analysis of the gridded perturbed AI data allows for the identification of
individual Arctic aerosol events, including a large BB aerosol plume that extended from Siberia over the Arctic Ocean
on 11 August 2019, shown in the Aqua MODIS true-color imagery found in Fig. 9b. The daily quarter-degree-gridded
perturbed AI data, shown in Fig. 9c, reveal a region of high perturbed AI data across northern Russia, with a plume
extending across the Arctic Ocean that closely matches the pattern found in the MODIS true-color imagery. The

number of quarter-degree grid boxes north of Greenland that contain daily average AI is much lower than in the
regions over and south of Greenland because of the reduced coverage of OMI AI data from each swath due to the row
anomaly. Only the last five OMI rows in each swath provide coverage north of Greenland, while rows 1 – 22 provide
additional coverage in the regions over and south of Greenland.

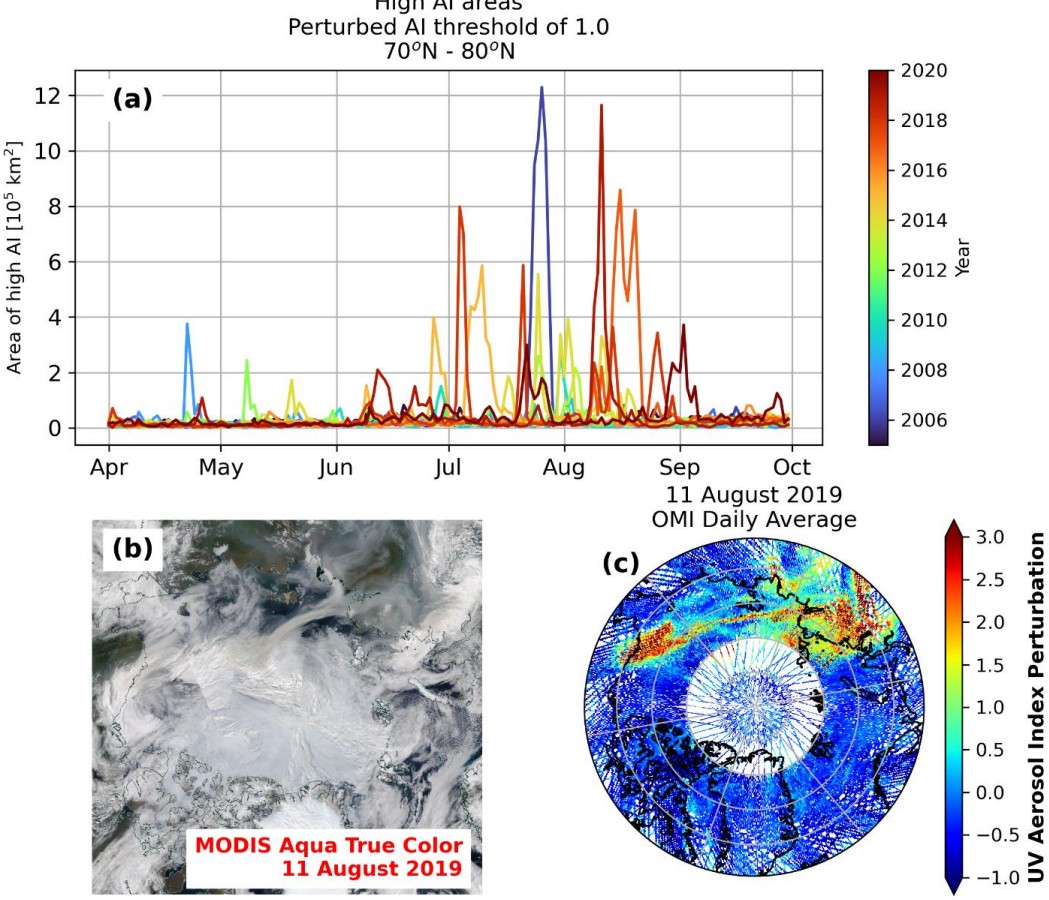

**Figure 9: a) Daily total area of quarter-degree lat/lon grid boxes between 70 °N and 80 °N with perturbed AI greater than
or equal to 1.0 between April 2005 and September 2020, with each time series colored by year. b) MODIS Aqua true color
composite (obtained from the NASA WorldView site at https://worldview.earthdata.nasa.gov/) imagery of a 11 August 2019
smoke plume extending from northern Siberia into the Arctic Ocean. c) Quarter-degree-gridded perturbed OMI AI data
for the 11 August 2019 BB aerosol event.**





When restricting the analysis to latitudes only north of 80º N, in latitudes and conditions that can largely only be sampled by OMI, a novel analysis into high Arctic aerosol events may be completed. Figure 10a shows similar daily time series of total high perturbed AI areas as in Fig. 9, but using only quarter-degree grid boxes north of 80º N. The number of large and small peaks is much smaller than in Fig. 9a, but as in Fig. 9a, the peak in aerosol events occurs in July and August. As indicated by the warm coloring of the small number of large peaks observed north of 80º N in

July and August, all high Arctic aerosol events occurred in the latter portion of the study period. An example of one of these high-Arctic BB aerosol events is shown in Fig. 10b, which shows Aqua MODIS true-color imagery of a plume extending over the Arctic sea ice near the North Pole; the plume is seen as a darkened region over the sea ice north of Greenland. In the daily quarter-degree-gridded perturbed AI data (Fig. 10c), the high Arctic plume signal can be seen in the same region as the plume in the MODIS true-color imagery, but because the high Arctic regions are only

sampled by 5 OMI sensor rows, the spatial resolution is much lower.

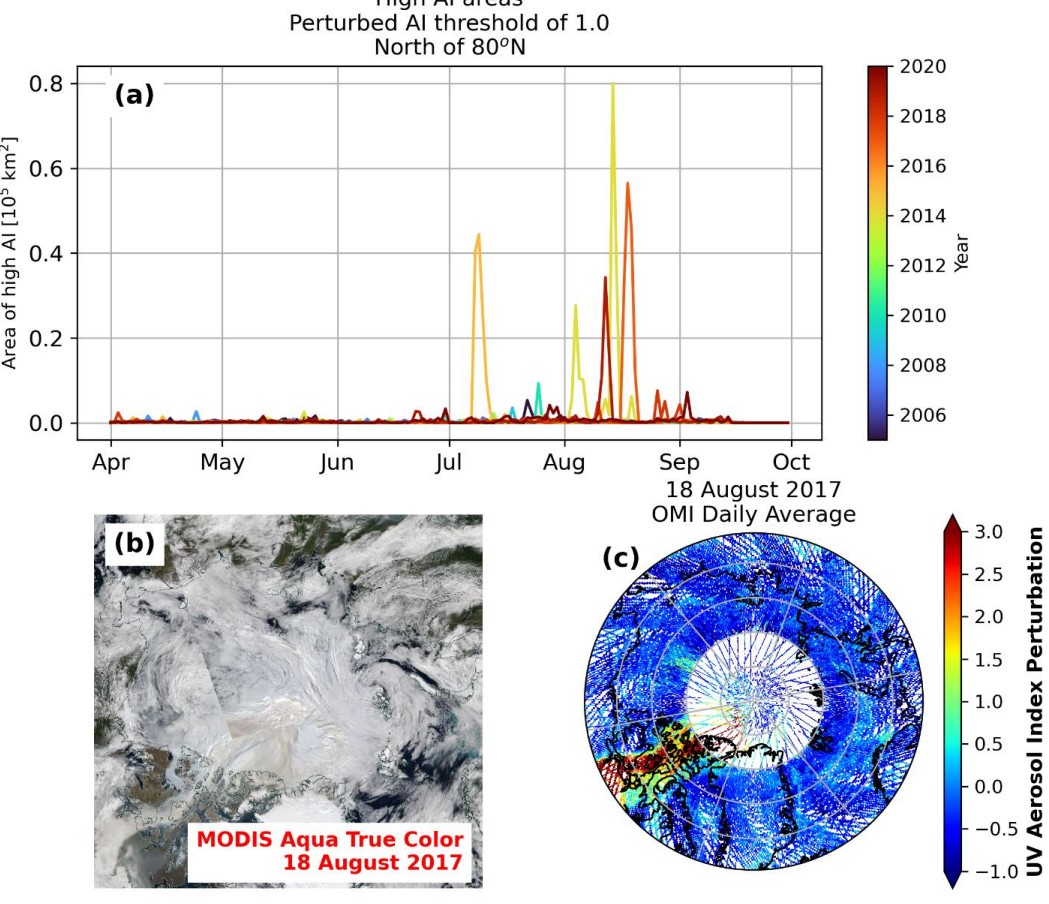

**Figure 10: As in Fig. 9, but for quarter degree lat/lon grid boxes north of 80 ºN. b) MODIS Aqua true color composite (obtained from the NASA WorldView site at https://worldview.earthdata.nasa.gov/) imagery of an 18 August 2017 smoke plume extending from northern Canada over the Arctic Ocean. c) Quarter-degree-gridded perturbed OMI AI data for the**

**18 August 2017 BB aerosol event.**





To investigate patterns in the daily AI area time series shown in Figs. 9 and 10, the number of peaks in the daily AI area time series area calculated for each year, with the areas of each peak binned into size ranges. Figure 11a shows the total time series of high (> 1.0) AI areas between $70^{\circ}$N and $80^{\circ}$N, with the 'x's indicating locations of peaks larger than $10^5$ km$^2$ in each year, while Fig. 11b shows the counts of high AI peaks in each size range per year. As indicated

350 by the number of 'x's in Fig. 11a and confirmed in the histogram shown in Fig. 11b, the number of high AI peaks per year in the latter half of the study period (2014 – 2020) is much larger than the number of high AI peaks per year in the earlier half of the study period (2005 – 2013). With the exception of 2016, every year between 2014 and 2020 saw at least 3 high AI area peaks, while no year between 2005 and 2013 saw more than 2 high AI area peaks. The size of the high AI events also increased throughout the study period from 2005 through 2020. Aside from the very large high

355 AI event in 2006, all high AI events in the first half of the study period were smaller than $5 * 10^5$ km$^2$, with only one reaching between $3 * 10^5$ km$^2$ and $5 * 10^5$ km$^2$ and reaching the second size bin. In the second half of the study period, many events occurred that were in the larger size bins. The high AI area peaks north of $80^{\circ}$N show similar results to the peaks north of $70^{\circ}$N, but with far fewer total events than in Fig. 11a and b. Figure 11c shows the daily total area of perturbed AI higher than 1.0 north of $80^{\circ}$N, with several large peaks found between 2014 and 2019. Unlike in Fig.

360 11a, in which there were several large BB aerosol events between $70^{\circ}$N and $80^{\circ}$N in the early half of the study period, there were no large-scale BB aerosol events in the first half of the study period north of $80^{\circ}$N; this is further visualized in the histogram of peak size ranges shown in Fig. 11d. It is worth noting that it is difficult to draw clear conclusions on high Arctic BB aerosol event trends from the results north of $80^{\circ}$N due to the small sample size of only 5 large-scale BB events, but we still report that all large BB aerosol events in the high Arctic (north of $80^{\circ}$N) between 2005

365 and 2020 occurred in the second half of the study period, between 2014 and 2020.



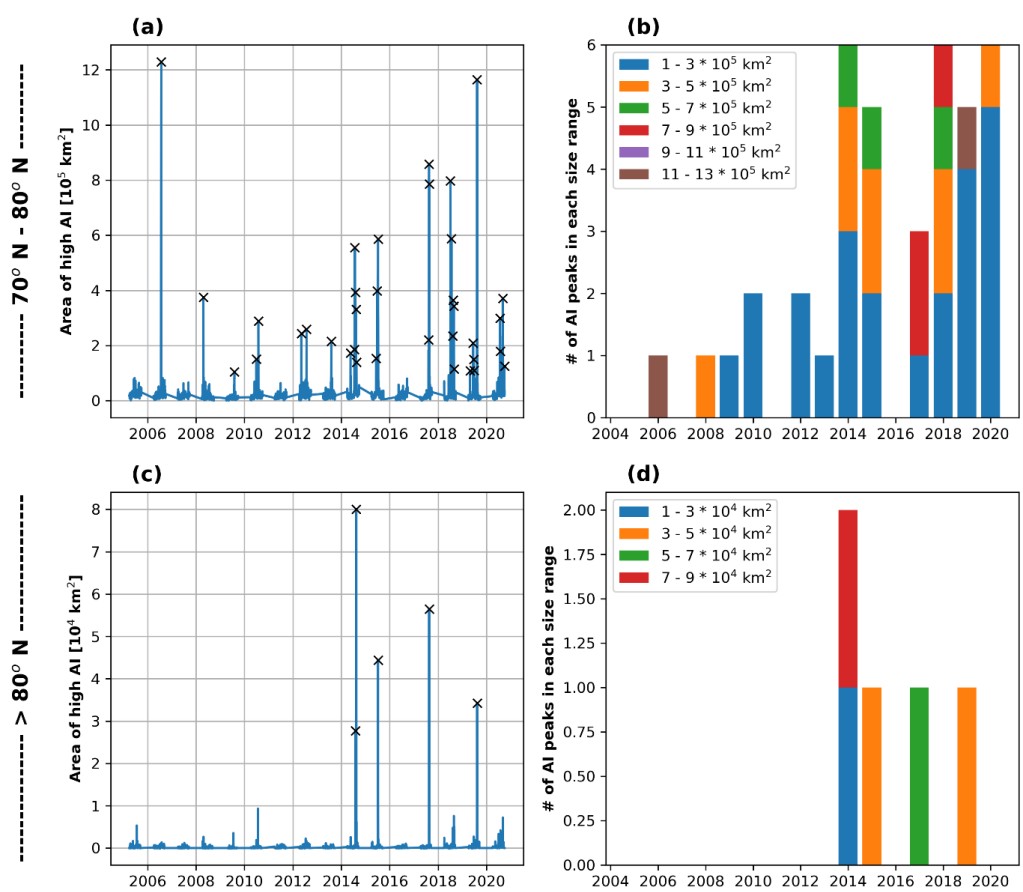

**Figure 11: a) Total time series of daily total areas of quarter-degree grid boxes with perturbed AI greater than 1.0 for grid boxes between 70º N and 80º N. The black 'x's denote peaks in the perturbed AI data. b) Yearly counts of peaks in each size range. c) as in a)., but for grid boxes north of 80º N. d) as in b), but for grid boxes north of 80º N. Note that the y axis values in panel c and the coverage bin areas in panel d are an order of magnitude smaller than in panels a and b.**

## 6. Conclusions

In this study, the feasibility of using OMI AI for studying spatiotemporal distributions of UV absorbing aerosols was investigated. Issues in OMI AI data over the Arctic regions were studied, and two quality controlled (QC) methods were developed for reducing bias and noise in OMI AI data for aerosol climate studies over the Arctic region. Lastly, QCed OMI AI data from both methods were used for studying the spatiotemporal variations of UV-absorbing aerosols over the Arctic region for the study period of 2005-2020. We found:



1. Non-trivial uncertainties exist in OMI AI data over the Arctic region. This includes unflagged row anomalies as well as systematic biases introduced by viewing geometry (e.g. higher bias is found for azimuth angle less than 100°) and certain surface types such as the "dry snow" surface type.

2. Two methods were developed for quality control of OMI AI data over the Arctic regions. The "screening method" was developed for using only the "best" OMI AI data from rows 56-60. This method provides unperturbed AI estimates, yet the data volume is very limited. Hopefully, some biases in OMI AI are rather systematic and are functions of observing conditions. Thus, the "perturbing method" was developed for estimating perturbations in OMI AI values from their climatological means. The climatological means of 15 years of OMI AI over the Arctic region were constructed as functions of surface conditions and viewing geometry and were found to contain systematic biases of OMI AI for given observing conditions.

3. Using QCed OMI AI data from the "screening" and the "perturbing" methods, spatiotemporal variations in OMI AI values were studied. We found decreasing AI values in spring and increasing AI over much of the Arctic region in the summer months, most notably in northern Russia and northern Canada in August, as well as decreasing AI over the Arctic Ocean north of Canada in June and July. Regional trends from both methods are largely consistent although some differences can be found that may due to the sampling differences between the two methods.

4. Using QCed data from the "perturbation method", we also studied extreme Arctic UV-absorbing aerosols events (perturbed AI > 1). We found increasing trends in the frequency and magnitude of high AI aerosol events over both the Arctic (70° N - 80° N) and high Arctic (> 80° N) regions. In particular, north of 80° N, no significant UV-absorbing aerosol events are found for the early part of the study period (2005-2013), yet a non-trivial frequency of significant UV-absorbing aerosol events are for the latter part of the study period (2014-2020), mostly in summer months, indicating intrusions of aerosol plumes near or above the North pole in recent years.

While the perturbed AI dataset generated for this study is designed for climatological and historical use, ongoing work is investigating the feasibility of directly assimilating the single-swath perturbed data into aerosol models for aerosol prediction over bright surfaces (Zhang et al., 2021).

**Code and data availability**

The OMI Level 2 UV aerosol index (UVAI) data used in this study to generate our monthly gridded analyses were obtained from the NASA Goddard Earth Sciences Data and Information Services Center (GES DISC) (https://disc.gsfc.nasa.gov/datasets/OMAERUV_003/summary, DOI:10.5067/Aura/OMI/DATA2004). Gridded QA-ed monthly OMI data from the screening and perturbing methods generated from this study and in netCDF4 format are included as a supplement.



**Author contributions**

J. Zhang and J. Reid designed the concept of the study. B. Sorenson implemented the study. S. Jaker processed the OMI data. All authors contributed to writing the manuscript.

**Competing interests**

The authors claim no competing interests.

**Financial support**

This project is supported by NASA Grant 80NSSC20K1260.

**Acknowledgements**

"We acknowledge the use of imagery from the NASA Worldview application (https://worldview.earthdata.nasa.gov), part of the NASA Earth Observing System Data and Information System (EOSDIS)."

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
