# Peer review of "OMI UV aerosol index data analysis over the Arctic region for future data assimilation and climate forcing applications"

_Atmospheric Chemistry and Physics, 2022_

## Author Comment (AC1)

**Comment**: *The identification of the seasonal "ring" features shown in Figure 1 are curious. Have these not been previously identified by the OMI team? And if not I suggest amplifying that this is an original finding as it would be a significant (i.e., important to know) aspect of the dataset that has not previously gotten scrutiny.*

Response: Thank you for the comment. To our knowledge, there has been no previous reporting of this "ring" feature over the Arctic region. Following your suggestion, we have amplified the identification of the ring feature to an original finding.

**Comment**: *It does not appear from the figures that you are excluding any OMI data for sub-pixel cloud contamination. You note on line 70 that the AI is calculated in clear and cloudy conditions, but there is a QA screening in the Level 2 that attempts to identify mainly clear pixels (QA=0) from cloud-contaminated (QA=1). Why is that QA consideration seemingly not used in this study?*

Response: Thank you for the suggestion. We do not include screening for sub-pixel cloud contamination here for two reasons. First, the impact of clouds on OMI aerosol index is relatively marginal compared to the impact of aerosol. For example, a change of AI value of around 1 is found for cloud optical depth increased from 2 to 20 in overcast conditions (Yu et al., 2012) and we expect smaller impacts from subpixel clouds. Secondly, cloud screening, and especially detection of sub-pixel clouds, is a challenging topic over the Arctic region from passive-based sensors due to bright and cold ice- and snow-covered surfaces. We suspect the OMI instrument experiences the same difficulty, as if we employ a strict cloud screening algorithm using the cloud-contamination flag in the L2 data (cloud fraction less than 0.2), much of the data over the Arctic sea ice is removed. We suspect that there are issues with the cloud screening method for OMI over the Arctic region (e.g. cloud free snow- and ice-covered regions are categorized as cloudy) and thus have not implemented the cloud screening flag as included in the OMI data.

Yu, H., Zhang. Y., Chin M., Liu Z., Omar A., Remer L. A. Yang Y., Yuan T. and Zhang J.: An Integrated Analysis of Aerosol above Clouds from A-Train Multi-sensor Measurements, Remote Sens. Environ., 121, 125–131, https://doi.org/10.1016/j.rse.2012.01.011, 2012.

**Comment**: *Line 93: "which seemingly latitude dependent" is not grammatically correct; please correct.*

Response: Thank you for the suggestion. We have changed to "which seem to exhibit a latitudinal dependence."

**Comment**: *Line 103: Please be more precise about the "northern end" of the swath that is meant in screening for unreported bad scan rows. The discussion of this issue seems to suggest*

*that it is a high latitude feature and not significant at lower latitudes (al a Figures 2a & b). Is that correct? Why would that be the case if it is indeed a physical obstruction of the sensor?*

Response: Thank you for the suggestion. As far as the "northern end" terminology, we mean the AI data over the Arctic region. We have removed the "northern end" terminology from the text and left the explanation as the OMI rows "over the Arctic region." In our analysis, we found that some of the unflagged rows exhibit anomalously high AI over the Arctic and report nominal AI values south of the Arctic region, while other unflagged rows are correctly flagged in non-Arctic regions but are not flagged in Arctic regions. We are unsure as to why these rows exhibit this behavior over the Arctic, but our analysis found that this behavior is present in the L2 OMI data.

Comment: *Section 3.2: In the discussion of other observing condition related defects in the dataset there is little discussion about limitations of the algorithm beyond the fact of different algorithms used over different surface types. You might consider Colarco et al. (2017) who identified other issues with the algorithm that are perhaps relevant here: there are biases in the OMI aerosol index visible to the extent that the surface pressure of the actual atmosphere differs from the static dataset assumed in the retrieval, and maybe more relevant to this discussion there are identified issues with the radiative transfer used in the retrieval algorithm having to do with the calculation of the Rayleigh atmosphere scattering over terrain where non-linear RT impacts were nevertheless linearly interpolated between two extreme pressures and that were manifest in bias in the AI. This could be relevant over topographically variable regions.*

Colarco, P. R., Gassó, S., Ahn, C., Buchard, V., Silva, A. M. da, and Torres, O.: Simulation of the Ozone Monitoring Instrument aerosol index using the NASA Goddard Earth Observing System aerosol reanalysis products, Atmos Meas Tech, 10, 4121–4134, https://doi.org/10.5194/amt-10-4121-2017, 2017.

Response: Thank you very much for the note. We have added discussion to Section 3.2 about the known surface pressure error- and RTM-induced AI biases from the Colarco et al (2017) paper.

Comment: *Line 203: The description of the climatology construction requires some further elaboration. If I want to know the AI at a particular latitude/longitude point, does your climatology tell me that? Is there a multi-dimensional histogram at each lat/lon point binned as described in SZA, VZA, etc? Is there no time dependence then in the climatological value at a given point? I think this just needs some additional clarification. (And how many bins of SZA, VZA, …?)*

Response: Thank you for the comment. As long as the viewing geometries and surface properties are provided for the desired lat/lon point, climatological AI can be computed. For example, if the climatology value for an OMI pixel is desired, the viewing geometry and surface properties associated with that OMI pixel are used to find the climatology value. There is no multi-dimensional histogram at each lat/lon point, as the climatology is not location-specific. There is a time dependence for the climatological values, as the climatology is constructed on a monthly

basis, as different months have different climatologies. We have added clarification to the methodology section to reflect the details of the climatology.

**Comment**: *It is not clear to me from Figure 7 that the conclusion there is no sensor drift is justified. Looking at the blue line are we supposed to conclude that it is stable after 2011? Because I see a lot variability in the maximum and minimum of the seasonal cycle (not to mention the high value after 2020 that is noted in the text). This analysis seems incomplete, or anyway not very convincing.*

Response: Thank you for your question. We want to ensure that there is no significant drift in the OMI sensor that could falsely generate trends over the Arctic, so we checked for significant drift here. We followed the reviewer's suggestion and modified Figure 7 to include the deseasonalized AI trends by removing seasonal patterns. There is still some slight variation in the monthly averaged AI from 2005 to 2020, with the increase in 2020 being the most prominent. Overall, however, these variations are very small, with most variations being less than 0.1 AI in magnitude, showing that there is no significant sensor drift. We also suspect the higher AI values from 2020 are real signals from Australian wildfires that spread smoke aerosols over large swaths of the Pacific Ocean, including our "remote ocean" zone, in 2020. We also added discussion accordingly.

**Comment**: *Line 295 and past: I don't understand this spatial sampling bias between the perturbed and screened assessments. Why would OMI rows 56-60 necessarily and systematically miss smoke events at high northern latitudes?*

Response: Thank you for the question. The perturbing method employs the use of all available sensor rows in any given month, while the screening method uses only the five rows 56 – 60, so even at the high northern latitudes, the addition of OMI rows 1 – 22 in the perturbed dataset can greatly affect the smoke events being sampled. Only rows 56 – 60 were selected for the screening method as those rows have high relative azimuth angles larger than 100° and are unaffected by row anomaly in the Arctic region through the entire study period. This reduces sampling-related biases (e.g. avoiding cases in which there are more rows in some years and fewer rows in other years) in the trend analysis. We are unsure what the reviewer means by OMI rows 56 – 60 "systematically miss smoke events at high northern latitudes." We assume smoke events and OMI rows are uncorrelated. We note how similar the perturbed and screened datasets are north of the 80° N circle in Figure 8; in that region, even the perturbed dataset only has coverage from OMI rows 56 – 60. Outside of that circle, however, the perturbed dataset has coverage from rows 1 – 22, which likely causes the differences between the June and July trends in the southern Arctic ocean.

**Comment**: *Line 302: What is the rationale for appealing to lower boundary condition issues? What does that even mean in this context?*

**Response**: Thank you for the comment. We appeal to lower boundary condition issues here because of the geographic differences in AI climatology and trend over the Greenland land mass versus the surrounding ocean water. We wouldn't expect meteorological sources to cause a clear difference in AI climatology and trend along the border of Greenland.

Suggestions for the figures:

**Comment**: Polar projection plots in Figures 1,2,3,5,6,8: Please put some lat/lon lines on the plots. In most cases you are referring in the text to specific latitude regions, so that would be helpful. Something like in Figures 9 & 10.

**Response**: Thank you for the suggestion. We have added lat/lon lines to Figures 1,2,3,5,6 and 8, and have modified the lines in Figures 9 & 10 to match the others.

**Comment**: The continuous color bar in Figure 9a should be replaced with a discrete one since the years are in discreet colors (I think).

**Response**: Thank you for the suggestion. The years are in discrete colors, so we have converted the continuous color bar to a discrete one.

**Comment**: Figure 11d: Suggest changing the y-axis label to show integer only labels since it is an integer quantity plotted.

**Response**: Thank you for the suggestion. We have changed the y-axis labels to show integers.

---

## Author Comment (AC2)

**Comment**: *As noted above, I think the framing of the discussion of the AI product itself is a bit wrong. Is it not the same sort of thing as an AOD retrieval and doesn't pretend to be. It's a semi-quantitative measure of the perturbation to UV reflectance coming (mostly) from absorbing aerosols. I think the analysis the authors have done here to transform it into something that can be looked at for trends is a good one. But I think the initial discussions of the AI product might make an unfamiliar reader feel like OMI AI is a bad data set that's full of artefacts. That's not the case: it's just if you want to use it in a meaningful way for quantitative climatology and trend analysis, you have to take all these extra steps, to account for these geometric/surface, etc dependencies baked in. I think this could be better articulated in the early part of the paper. See for example Torres et al (1998), Hsu et al (1999) which discussed these issues (talking about both aerosol index and AOD). The authors write "semi-quantitative" in the Abstract, which is at least something, but I think this needs to be given more space in the paper itself.*

Response: Thank you for the suggestion. We have added text to the early portions of the paper to reflect this.

**Comment**: *The trend analysis (section 4.3) is done in a common way: do ordinary least squares linear regression on the time series of perturbations, and do a T-test to identify grid points where the p-value is below 0.05. The results are framed in terms of this linear AI perturbation gradient and the locations of low p-value (which correspond to points for which, if there is truly no trend in the time series, the chances of observing an apparent linear trend at least this large are lower than 5% - at least I believe this is the correct interpretation). This is a common way of doing things but has a few issues which should be acknowledged. One is that my doing these tests pointwise on a map we are not doing single hypothesis testing but rather multiple; further, since the source data are highly spatially correlated, `noise' in the fit can be correlated as well, leading to blobs of apparent significant trend which may or may not be real but look realistic because they are spatially coherent. Wilks (2016) has an important discussion of this and some suggestions (references therein) to use a dynamic p-value to control on the false discovery rate instead. Another approach (which I personally prefer) is not to focus on significance but rather look at estimated trends and uncertainties on those estimates (which should be provided by whatever linear regression routine is used). On reason is because `insignificant' is not one thing: if you have an `insignificant' trend with a low uncertainty on the trend estimate you can fairly confidently rule out there being a large trend; if you have an `insignificant' trend with a high uncertainty on the trend estimate then any true trend might be large or small (and we might not know the sign). It is not clear from the analysis done how much of the `insignificant' trend areas might fall into each sub-category. I suggest the authors try looking at maps of AI trend and AI trend uncertainty and see if they can make some assessment of this (it doesn't need to be shown in the paper, just some statements of what is the typical level of precision on the trend estimates in various cases and therefore where we can/can't rule out some missed important trend).*

Response: Thank you for the suggestion. We followed your suggestion and compared maps of standard error of the trend (slope) to the trend maps (included as a supplement). We have added

appropriate discussion in Sections 4.3 and 5.1, and have included the following figure, which compares the trends and trend standard errors, as a supplement to the paper.

[Figure]

Figure S1: April (first and top row), May (second row), June (third row), July (fourth row), August (fifth row), and September (sixth and bottom row) monthly trend of screened OMI AI data (first column) and perturbed AI data (second column), as well as the standard error of the screened (third column) and perturbed (fourth column) AI trends. Trends are calculated between 2005 and 2020. The dotted regions in the left two columns denote trends that are statistically significant at the 95% confidence level.

**Comment**: *A further issue that I think should be mentioned is that trends on a time series of monthly mean perturbations might not make sense if trends are driven by changes in the number of extrema rather than the baseline AI (since we know aerosol distributions tend to be skewed with a long tail). Plus, it's not clear that a linear model is appropriate for the same reason. This ties into the above as significance testing and uncertainties are predicated on the assumed model. I do appreciate that the analysis was done separately for each month (since trends can differ between months). Plus, the authors do not infer too much from the quantitative AI trends – more when and where they are happening – which alleviates those quantitative concerns a bit. But, the fact that choice of model for trend construction is important should be acknowledged.*

Response: Thank you for the suggestion. We added discussion emphasizing the importance of method choice for trend analysis to the text. We also mentioned that if trends are driven by a few extreme events, the trend may not make sense.

**Comment**: *The acronym QC should be defined at first use (I know what it means but some readers might not).*

Response: Thank you for the note. We have added a definition of the QC acronym.

**Comment**: *I was a bit surprised there was no mention of e.g. TropOMI here. Not expecting it to be included in the analysis given it was launched in 2017 but it could be useful to point to its advantages for this type of work over OMI (e.g. spatial resolution, no row anomaly) for the future. Likewise OMI's advantages over TOMS, etc (again spatial resolution) could be mentioned. I don't know that much discussion is needed but a mention wouldn't be amiss. Not sure the geo spectrometers need mentioning, though, since they won't observe the Arctic.*

Response: Thank you for the suggestion. We have added discussion to the introduction about OMI's spatial resolution advantage over TOMS, and about TROPOMI's benefits over OMI for future Arctic aerosol analyses in the conclusion section. We have elected to leave the geo spectrometers out of the discussion, since, as you mentioned, they don't observe the Arctic.